# Whole Brain and Corpus Callosum Fractional Anisotropy Differences in Patients with Cognitive Impairment

**DOI:** 10.3390/diagnostics13243679

**Published:** 2023-12-16

**Authors:** Kalvis Kaļva, Nauris Zdanovskis, Kristīne Šneidere, Andrejs Kostiks, Guntis Karelis, Ardis Platkājis, Ainārs Stepens

**Affiliations:** 1Department of Radiology, Riga Stradins University, LV-1007 Riga, Latvia; kalviskalva@gmail.com (K.K.);; 2Department of Radiology, Riga East Clinical University Hospital, LV-1038 Riga, Latvia; 3Military Medicine Research and Study Centre, Riga Stradins University, LV-1007 Riga, Latvia; 4Department of Health Psychology and Paedagogy, Riga Stradins University, LV-1007 Riga, Latvia; 5Department of Neurology and Neurosurgery, Riga East University Hospital, LV-1038 Riga, Latvia; andrejs.kostiks@gmail.com (A.K.);; 6Department of Infectology, Riga Stradins University, LV-1007 Riga, Latvia

**Keywords:** diffusion tensor imaging, Montreal Cognitive Assessment, fractional anisotropy, cognitive impairment, neuroradiology, cognition, mild cognitive impairment, dementia

## Abstract

Diffusion tensor imaging (DTI) is an MRI analysis method that could help assess cognitive impairment (CI) in the ageing population more accurately. In this research, we evaluated fractional anisotropy (FA) of whole brain (WB) and corpus callosum (CC) in patients with normal cognition (NC), mild cognitive impairment (MCI), and moderate/severe cognitive impairment (SCI). In total, 41 participants were included in a cross-sectional study and divided into groups based on Montreal Cognitive Assessment (MoCA) scores (NC group, nine participants, MCI group, sixteen participants, and SCI group, sixteen participants). All participants underwent an MRI examination that included a DTI sequence. FA values between the groups were assessed by analysing FA value and age normative percentile. We did not find statistically significant differences between the groups when analysing CC FA values. Both approaches showed statistically significant differences in WB FA values between the MCI-SCI and MCI-NC groups, where the MCI group participants showed the highest mean FA and highest mean FA normative percentile results in WB.

## 1. Introduction

Cognitive impairment is defined as a disruption in cognitive functions, such as memory, attention, language, problem solving, and executive functioning [1]. If cognitive impairment reaches a point where it severely impacts a person’s daily functioning and prevents their active engagement in social interactions, it suggests a potential transition to dementia. Dementia is a neurodegenerative condition characterised by a significant level of cognitive impairment that hampers an individual’s ability to actively participate in social and work life [2]. Cognitively impaired individuals often require increased support and care from their families, caregivers, and healthcare providers, leading to higher healthcare costs and reduced productivity in the workforce [3,4].

The current diagnostic methods for cognitive impairment primarily rely on clinical assessments, such as neuropsychological testing and brain imaging techniques, like magnetic resonance imaging. However, these methods have certain limitations and may not accurately reflect the extent of cognitive impairment [5].

Diffuse tensor imaging (DTI) is a magnetic resonance imaging-based technique that allows for the exploration of the integrity of white matter microstructure by measuring the degree and directionality of water diffusion, and the overall magnitude of diffusion could be measured, as indicated by the Apparent Diffusion Coefficient (ADC) [6]. Fractional anisotropy is a metric used in diffusion tensor imaging to assess the microstructural integrity of white matter fibre bundles in the brain. It measures the degree and directionality of diffusion within white matter tracts, providing information about the organisation and integrity of these tracts [7].

This type of examination is important for several reasons. First of all, this technique provides valuable information about the microstructure of brain tissue and can be used to evaluate white matter abnormalities [8,9]. 

The use of DTI has proven to be beneficial in the evaluation of multiple sclerosis (MS), as it has the capability to offer quantitative parameters, like FA and mean diffusivity (MD) [10]. The method has demonstrated promise in visualising and quantifying disruptions in connectivity and assessing variations in normal-appearing white matter in patients with MS [11]. It has also been shown to differentiate between various types of brain tumours, including glioblastoma and cerebral metastases, by visualising white matter tracts and providing information on tumour infiltration into surrounding brain tissue [12,13,14]. For example, DTI-derived metrics have been used to distinguish glioblastoma multiforme from normal brains, and DTI-guided radiotherapy has been suggested as an advanced approach [15,16]. Furthermore, it has also shown promise in predicting patterns of glioma recurrence, enabling improved customisation of tumour management and stratification for randomised controlled trials [17]. The method has also been widely employed to study the effects of various conditions on brain microstructural abnormalities, including traumatic brain injury [18], type 2 diabetes with mild cognitive impairment [19], and myotonic dystrophy type I [20]. Additionally, DTI has been used to study the frontal lobe white matter in children with autism spectrum disorder (ASD), revealing variances in neural pathways compared to typically developing children [21]. In a study by Sivaswamy et al., DTI was utilised to examine specific brain pathways, such as the cerebellar pathways, and has identified irregularities in children with ASD [22].

One of the key applications of DTI analysis is fibre tracking in the brain. By combining DTI with functional MRI, researchers can gain insights into the connectivity of different brain regions [9]. This is particularly important for understanding the neural networks and pathways involved in various brain functions and processes.

FA alterations also may be region-specific and associated with respective clinical manifestations. For instance, studies have demonstrated that changes in FA in specific white matter tracts are associated with cognitive impairment in memory clinic patients with vascular brain injury [23]. Furthermore, FA alterations in the corpus callosum have been linked to chronic visual neglect [24], autism spectrum disorder [25], stroke, multiple sclerosis, dyslexia, and schizophrenia [9]. They can also be sensitive markers of brain microstructural alterations, especially after mild traumatic brain injury [26]. By examining the diffusion properties of water molecules in affected brain regions, DTI can provide insights into the structural changes associated with these diseases. This information can be used for diagnostic purposes, monitoring disease progression, and evaluating treatment efficacy [8]. In addition to its clinical applications, DTI analysis is also becoming part of routine clinical protocols. Its ability to provide detailed information about tissue microstructure and connectivity makes it a valuable tool for clinicians in various medical fields [9]. DTI analysis can aid in the diagnosis and management of brain diseases and disorders, contributing to improved patient care and outcomes [9]. 

DTI has been used to diagnose intracranial pyogenic infections, masses, trauma, and vasogenic versus cytotoxic edema [27]. It has also been applied in the investigation of cerebral ischemia, brain maturation, traumatic brain injury, epilepsy, multiple sclerosis, Alzheimer’s disease, brain tumours, and metabolic disorders [27]. By examining WB FA, researchers could gain a comprehensive understanding of the structural changes associated with these conditions.

It can be assumed that WB FA analysis allows for a comprehensive assessment of white matter integrity throughout the brain. By examining FA values across the entire brain, researchers can identify global patterns of white matter alterations. This approach has been used to investigate the effects of housing quality and behaviour on brain health and anxiety [28]. What is interesting is that whole brain FA analysis has been employed to study the time course of Wallerian degeneration after ischemic stroke [29].

However, there are certain limitations and challenges associated with DTI analysis that make it less suitable for certain applications. Limitations include different acquisition parameters (single-shell vs. multi-shell, number of directions), different preprocessing and post-processing techniques, and software differences [30]. Other limitations include DTI’s inability to resolve multiple fibre orientations within a single voxel. The tensor model used in DTI assumes a single dominant fibre orientation, which means that it cannot accurately represent complex fibre configurations such as fibre crossing, bending, or twisting [9]. All these limitations can lead to inaccurate interpretation of the data and may result in misleading conclusions.

In a study by Porcu et al., the fractional anisotropy of healthy participants was analysed, revealing that as age increases, FA decreases. There were no significant differences between FA and gender. Higher FA values were associated with increased activity in specific brain regions, including those within the default mode network. This suggests that higher FA may be linked to enhanced neural activity during resting states. The study also found a stronger correlation between certain brain regions, particularly those that are part of the limbic system. This suggests that FA may play a role in influencing the connectivity and networking of brain regions, particularly within the limbic system [31].

Furthermore, DTI analysis may not be suitable for all types of tissues or diseases. While DTI has been successfully applied in studying brain white matter and demonstrating abnormalities in diseases, such as stroke, multiple sclerosis, dyslexia, and schizophrenia [9], its applicability to other tissues or diseases may be limited. For example, a study evaluating acute anterior ischemic optic neuropathy found that DTI parameters did not show significant differences between affected and unaffected eyes [32]. This suggests that DTI may not be sensitive enough to detect subtle changes in certain tissues or diseases.

To sum up the introduction, recent advancements and ongoing studies in DTI research continue to expand our understanding of its applications in cognitive impairment. Continuous developments in acquisition parameters, processing techniques, and analytical models are addressing some of the limitations previously mentioned, enhancing DTI’s utility in neuroimaging research. For instance, Tax et al. summarised preprocessing steps, limitations, and future steps, focusing on what is new in the post-Human Connectome Project era. They highlighted the emergence of databases and simulations specifically targeted at evaluating preprocessing steps and efforts that perform automated quality control and quantitatively and qualitatively compared preprocessing steps. They also discussed practical considerations and provided insights into what is next in DTI analysis [30].

Our study’s objective was to compare WB FA and CC FA results between the three groups (normal cognition (NC), mild cognitive impairment (MCI), and severe cognitive impairment (SCI)), which were divided based on MoCA test results. 

## 2. Materials and Methods

In total, 41 participants were included in the cross-sectional study, who were then divided into 3 groups according to Montreal Cognitive Assessment (MoCA) scores [33,34]:Normal cognition (NC) group (participants with MoCA scores ≥ 26);Mild cognitive impairment (MCI) group (participants with MoCA ≥ 20 and ≤25);Severe cognitive impairment (SCI) group (participants with MoCA ≤ 19.

There were 9 participants in the NC group (M age 65, SD 11.456, youngest 44 years, oldest 77 years, mean MoCA score 28.222, SD 1.093, lowest score 27.000, highest score 30.000), 16 patients in the MCI group (mean age 69.563, SD 7.330, youngest 57 years, oldest 80 years, mean MoCA score 23.313, SD 1.662, lowest score 20.000, highest score 25.000), and 16 patients in the SCI group (mean age 75.938, SD 10.491, youngest 62, oldest 96, mean MoCA score 10.750, SD 4.946, lowest score 4.000, highest score 18.000). Research participant demographic data, gender, and MoCA scores between the groups can be seen in Table 1.

A chi-square test on gender was conducted, and it determined that there were no statistically significant differences between the groups (X^2^ = 2.259, *p* = 0.323). Similarly, no group differences were found in age (X^2^ = 60.575, *p* = 0.052) using a Kruskall–Wallis test; nevertheless, statistically significant differences between the three groups were identified in MoCA scores (X^2^ = 82.000, *p* < 0.001).

### 2.1. Selection of Participants

The participants included in our study were referred to a neurologist based on their subjective complaints of cognitive impairment or suspected cognitive impairment based on primary physician assessment.

Participants were excluded from this study if they had clinically significant neurological or psychiatric disorders (such as a history of tumours, severe strokes, vascular malformations, major depression, Parkinson’s disease, schizophrenic disorders, bipolar disorders, maniacal states, etc.), as well as a history of drug or alcohol abuse. 

The neurologist who participated in our research is a board-certified professional with expertise in diagnosing and managing cognitive impairment.

No other clinically significant abnormalities were detected on the MRI scans in patients enrolled in this study. None of the participants had signs of cerebral amyloid angiopathy, more than 4 microbleeds, intra-/extra-axial tumours, vascular malformations, or signs of other neurodegenerative diseases. From the available clinical records, none of the participants had uncontrolled hypertension, diabetes mellitus, or clinically verified depression. All participants were university graduates with at least 16 years of education. The cognitive testing and MRI data were obtained in the time period from January 2020 to December 2022.

In our research work, ASL and fMRI sequences were not performed.

### 2.2. MRI Acquisition Protocol and Fractional Anisotropy Calculation

All patients underwent an MRI scanning protocol based on the Alzheimer’s Disease Neuroradiology Initiative, which included:Three-dimensional T1 SPGR (technical parameters—flip angle 11, TE min full, TI 400, FOV 25.6, layer thickness 1 mm);Three-dimensional FLAIR (technical parameters—TE 119, TR 4800, TI 1473, echo 182, FOV 25.6, layer thickness 1.2 mm);High-resolution hippocampal structure assessment sequence (technical parameters—flip angle 122, TE 50, Echo 1, TR 8020, FOV 17.5, layer thickness 2, coronal direction perpendicular to the hippocampus);DTI (technical parameters—32 directions, b = 0 and 1000 s/mm^2^, diffusion direction—tensor, FOV 23.2, layer thickness 2 mm, TE 100);SWI (technical parameters—flip angle 15, TE 22.5, TR 34.7, slice thickness 3 mm);DWI (technical parameters—b = 0, 1000, and synthetic 2000 s/mm^2^, flip angle 90, TE 76.0, TR 9852.0, slice thickness 3 mm).

The DTI evaluation was performed using the Icometrix DTI software (icobrain tbi report) package, which evaluates fractional anisotropy in the whole brain, corpus callosum, and other specific brain tracts. The examinations were performed on a single 3T MRI machine, images were preprocessed, and DTI data were processed to acquire FA values for the whole brain and corpus callosum (see Figure 1).

Performed steps for processing included eddy current correction with affine registration, the use of Tractseg [35] for creating a binary tract mask, and then computing the mean FA from iteratively reweighted linear least squares in the tract mask [36]. Further, FA maps and distributions are calculated and extracted from the region of interest (whole brain and corpus callosum) [37].

After all steps have been completed, the resulting FA data are standardised, harmonised, and compared to age- and gender-normative data obtained from a healthy population database that consists of 918 MR studies from 788 unique patients aged from 18 to 86 years, acquired on different scanners and equally distributed over different age groups [37,38,39].

In our study, the analysis of WB FA was specifically concentrated on white matter, using TractSeg for accurate segmentation. This focused approach excludes grey matter and cerebrospinal fluid, allowing us to precisely assess white matter integrity. Corpus callosum FA was extracted based on the TractSeg segmentation protocol with further FA calculation as described above.

In our research, focusing on both the whole brain and the corpus callosum, we specifically analysed two key variables: the fractional anisotropy (FA) value and the normative percentile, which take into account age and gender-based norms.

### 2.3. Statistical Analysis

JASP 0.17.3 was used for statistical analysis (Eric-Jan Wagenmakers, Amsterdam, The Netherlands) [40]. Statistical analysis included descriptive statistics, a chi-square test, a Kruskal–Wallis test, and Dunn’s post hoc analysis of study results. 

Descriptive statistics were used to estimate general variables and differences between groups. A chi square test was used to determine the association between categorical variables. A Kruskall–Wallis test was used to evaluate statistically significant differences between the 3 groups, and if there were statistically significant differences, Dunn’s post hoc test was utilized with additional Bonferroni and Holm corrections.

## 3. Results

### 3.1. Whole Brain Fractional Anisotropy

Using a Kruskal–Wallis test, we identified statistically significant differences in whole brain fractional anisotropy (WB FA) (H (2) = 9.311, *p* = 0.010) (see Figure 2).

By performing Dunn’s post hoc test, statistically significant differences were found between the SCI-MCI group (*p* = 0.007, after Bonferoni and Holm correction statistical significance was maintained) and the MCI-NC group (*p* = 0.016, after Bonferoni and Holm correction statistical significance was maintained) (see Table 2).

By performing a Kruskal–Wallis test for WB FA, normative percentile differences were found to be statistically significant between groups (H (2) = 11.614, *p* = 0.003) (see Figure 3).

By performing Dunn’s post hoc test, statistically significant differences were found between the SCI-MCI group (*p* = 0.018, but after Bonferoni correction there were no statistically significant differences) and the MCI-NC group (*p* = 0.001, after Bonferoni and Holm correction statistical significance was maintained) (see Table 3).

### 3.2. Corpus Callosum Fractional Anisotropy

The Kruskal–Wallis test did not indicate statistically significant differences between the study groups (H (2) = 0.565, *p* = 0.754) (see Figure 4).

The highest mean corpus callosum (CC) FA was found in the MCI patient group (the mean value in the group was 0.628), and the lowest was in the SCI group (the mean value in the group was 0.617).

Similarly, as in CC FA values, for the CC FA normative percentile there were no statistically significant differences between our study groups when the Kruskal–Wallis test was performed (H (2) = 0.376, *p* = 0.829) (see Figure 5).

The highest mean CC FA normative percentiles were in the MCI group (the mean value in the group was 61.962), and the lowest mean percentiles were in the SCI group (the mean value in the group was 55.262). 

## 4. Discussion

While the analysis of diffusion tensor imaging fractional anisotropy (DTI FA) data in patients with cognitive impairment provides some insights into the microstructural alterations within white matter tracts of the brain, in our study, when analysing whole brain FA and CC FA, we did not observe substantial differences in FA values among NC-SCI. We observed that FA values are higher in the MCI group compared to the NC and SCI groups. While this outcome may initially appear inconclusive, it offers valuable insights into the complex relationship between cognitive impairment and microstructural white matter alterations.

Studies have shown that reduced FA values are associated with conditions, such as traumatic brain injury [41], multiple sclerosis [42], and Parkinson’s disease [43], and conversely, increased FA values have been observed in individuals born preterm [44] and in adolescents with severe perinatal brain injury [44]. These findings suggest that changes in FA can reflect both pathological and compensatory processes in the brain. In the context of our findings, lower whole brain FA results in dementia, and severe cognitive impairment is a common finding, emphasising the degenerative changes that often accompany cognitive decline [45,46,47].

Contrary to the previous statement, in our study, whole brain FA values were highest in the MCI group. There could be several explanations. 

In whole brain FA analyses, the intrinsic variability of individual white matter tracts is disregarded. This methodological limitation can result in compromised sensitivity and specificity when interpreting alterations in global FA metrics. Therefore, such an approach may not be optimally suited for capturing nuanced neuroanatomical changes that affect cognition. 

Whole brain FA value changes during a lifespan is a non-linear value, where several white matter tracts in the brain obtain peak FA at a different age, i.e., the highest FA of cingulum is observed around the age of 43, and the highest FA of inferior fronto-occipital fasciculus is observed around the age of of 25 [48,49]. Barnea-Goraly et al. also observed age-related changes in white matter FA values in four brain pathways, including the corpus callosum and white matter tracts within the basal ganglia [50]. However, there is also evidence suggesting a linear decrease in white matter connectivity with age. Research by Webb et al. found that white matter connectivity in the optic radiation exhibited a linear decrease across the lifespan, indicating age-related degradation of white matter and its impact on visual executive functions [51]. In addition to age-related changes, vascular burdens and genetic factors may also contribute to white matter microstructural decline. A study by Williams et al. found that higher cholesterol levels were associated with a poorer white matter microstructure in cognitively normal older adults, suggesting that vascular diseases can impact white matter integrity [52].

Several papers indicate widespread variations of FA changes where in longitudinal studies some participants observe an increase in FA and some observe a decrease in FA in the same regions. In a study by Lebel and Beaulieu, the majority of participants exhibited increases in FA in association fibres. Notably, these specific fibres, including the superior and inferior fronto-occipital, inferior longitudinal, and cingulum, continued to show FA increases between scans, even among individuals aged 19–25 and 22–32 years. The study measured 10 brain tract FA—the genu, body, and splenium of the corpus callosum, corticospinal tracts, superior and inferior longitudinal fasciculus, superior and inferior fronto-occipital fasciculus, uncinate fasciculus, and cingulum—and it stated that a considerable proportion of the participants, ranging from 20% to 30%, experienced a decrease in FA. These changes in FA are typically interpreted as unfavourable or detrimental in the context of ageing in the elderly population [53].

Moreover, Pareek et al. and Rathee et al. did not observe age-related changes in whole brain FA when analysing patients in three age groups: young adults (20–40 years), middle age (41–60 years), and old age (61–85) [54,55]. However, there is evidence that age-related changes in grey matter thickness and FA are driven, in part, by a common biological mechanism related to changes in cerebral myelination. This suggests that age-related changes in whole brain FA may be influenced by changes in myelination in the cerebral white matter [56]. Furthermore, Wright et al. compared age-related decline in whole brain FA values in patients with schizophrenia and normal controls and found that the decline was approximately twice as fast in patients compared to the controls. This indicates that factors, such as neurological disorders, can also impact age-related changes in whole brain FA.

Despite not finding any statistically significant age differences between the study groups, it is worth noting that the participants in the NC group were younger. This finding further undermines the reliability of using WB FA as a biomarker for early cognitive impairment.

CC plays a crucial role in facilitating interhemispheric communication and coordination. CC FA results provide insights into the organisation of the white matter fibres connecting the left and right cerebral hemispheres. In our study, we compared the whole corpus callosum FA value, but other research suggests that different segments of the CC, such as the genu, body, and splenium, can be separately analysed to assess FA alterations in specific regions. For example, previous research has shown reduced FA in the genu of the CC in individuals with Alzheimer’s disease and decreased FA in the splenium in individuals born preterm [44]. 

These findings suggest that different regions of the corpus callosum may be selectively affected in different neurological conditions. Therefore, further research could be focused on specific corpus callosum regions. 

It is also crucial to acknowledge the limitations of our study. One possible explanation for the lack of significant differences in FA among the study groups is the heterogeneity of cognitive impairment within each group. Cognitive impairment is a complex and multifactorial condition, influenced by various etiological factors, including age, genetics, and comorbidities [57]. It is plausible that the variability within each group masked any potential differences in FA that may exist between them. Future studies should consider controlling for these confounding factors to obtain a more accurate assessment of the relationship between FA and cognitive impairment severity [58]. It is worth noting that our sample size was relatively small, which may have limited the statistical power to detect significant differences. A larger sample size would provide more robust results and increase the generalisability of our findings. Additionally, the inclusion of longitudinal data would allow for the examination of FA changes over time and provide insights into the dynamic nature of cognitive impairment. Potential confounding variables that were not detected on initial screening are also significant factors that influenced the study results [59]. Additionally, highly educated individuals with actual cognitive decline may perform well in these tasks and thus mask their decline [60]. 

It is important to mention that DTI is limited in its ability to resolve multiple fibre orientations within a single voxel. This can be problematic in regions where multiple fibre bundles intersect or cross each other, leading to partial volume effects and difficulties in accurately characterising complex fibre architecture [59]. Noise in DTI data can have a significant impact on the accuracy and reproducibility of the derived measurements. Studies have shown that noise can affect diffusion anisotropy indices, mean diffusivity, and principal eigenvector measurements [61]. In particular, the estimation process used in DTI has been found to be sensitive to noise [62]. The effect of noise on DTI data is complex and can lead to a systematic distortion of the tensor, which depends on the signal-to-noise ratio (SNR) [63]. DTI assumes Gaussian diffusion, which may not accurately represent diffusion behaviour in regions with restricted diffusion, such as areas with crossing fibres or complex tissue microstructures. Generalised diffusion tensor imaging (GDTI) methods have been proposed to address this limitation and capture non-Gaussian diffusion behaviour [64]. 

These limitations may have contributed to our inability to detect significant FA differences. Therefore, further research with larger sample sizes, longitudinal designs, and a more comprehensive assessment of clinical variables is necessary to evaluate DTI FA use in cognitive impairment diagnostics. 

Also, we used Icometrix icobrain tbi report, which uses specific preprocessing algorithms that may or may not be used in other research papers analysing FA. Factors such as fibre crossing, partial volume effects, and tissue heterogeneity can complicate the interpretation of diffusion metrics and their relationship to underlying tissue properties [65].

The lack of significant differences in FA between study groups suggests that FA may not be a reliable standalone biomarker for assessing the severity of cognitive impairment. This raises questions about the utility of FA in clinical settings. It is possible that FA alone does not capture the complexity and heterogeneity of cognitive impairment and that a multimodal approach incorporating other neuroimaging measures, such as cortical thickness or functional connectivity, may provide a more comprehensive assessment. Analysing specific tracts can offer a more detailed understanding of the structural changes associated with varying degrees of cognitive impairment. Future studies should consider integrating multiple imaging modalities to gain a more holistic understanding of the neural correlates of cognitive impairment.

Our objective was to compare WB FA and CC FA results between the study groups that were divided based on MoCA test results. We observed statistically significant differences between the MCI-SCI and NC-MCI groups.

## 5. Conclusions

In our study, we observed statistically significant differences in WB FA between the SCI-MCI and NC-MCI groups, where the MCI group participants had the highest mean FA and highest mean FA normative percentile results in WB. We did not observe statistically significant differences between the groups by analysing CC FA and CC FA normative percentiles.

Therefore, future research could prioritise tract-specific analyses over whole brain evaluation to achieve a more specific and potentially clinically relevant understanding of FA variations in cognitive impairment.

## Figures and Tables

**Figure 1 diagnostics-13-03679-f001:**
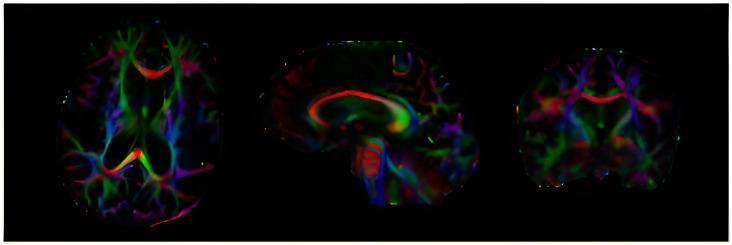
DTI examination with colour-coded fibre pathways in axial, sagittal, and coronal planes.

**Figure 2 diagnostics-13-03679-f002:**
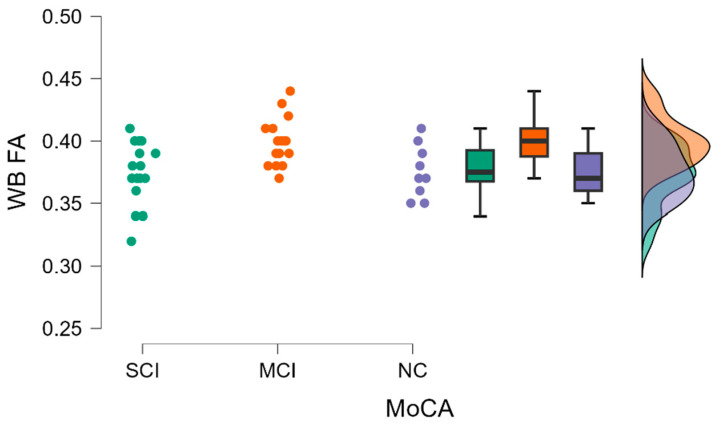
WB FA value comparisons between (from left to right) SCI, MCI, and NC with data distribution in each group.

**Figure 3 diagnostics-13-03679-f003:**
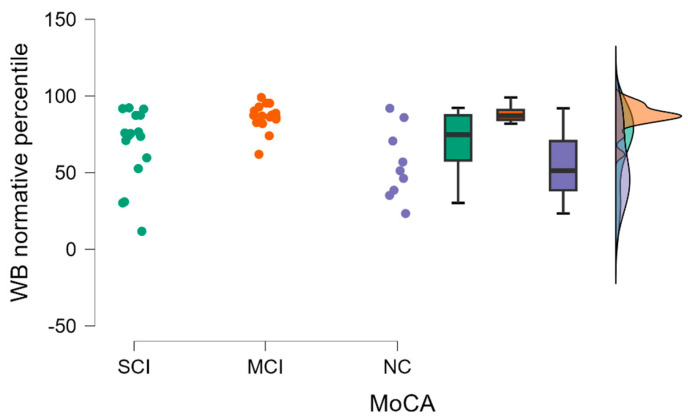
WB FA normative percentile comparisons between (from left to right) SCI, MCI, and NC with data distribution in each group.

**Figure 4 diagnostics-13-03679-f004:**
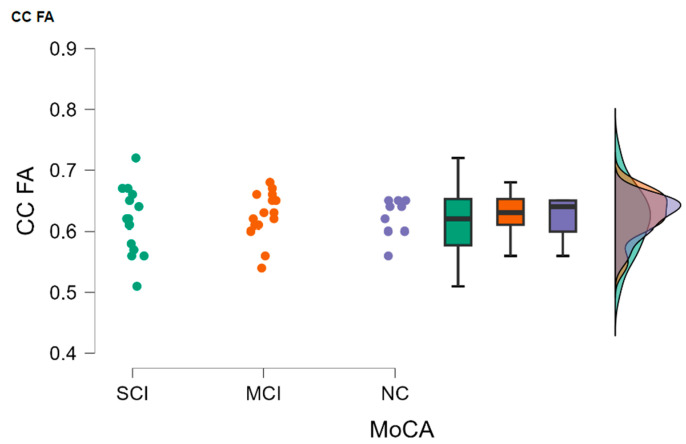
CC FA comparisons between (from left to right) SCI, MCI, and NC with data distribution in each group.

**Figure 5 diagnostics-13-03679-f005:**
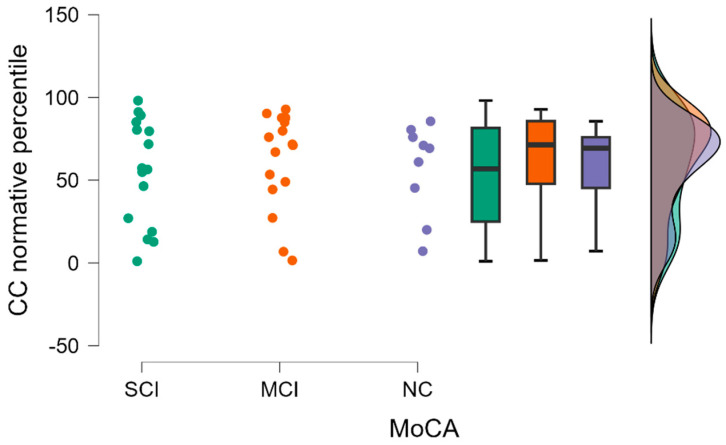
CC FA normative percentile comparisons between (from left to right) SCI, MCI, and NC with data distribution in each group.

**Table 1 diagnostics-13-03679-t001:** Research population and division according to MoCA score.

	Gender (F:M)			Age			MoCA		
	NC	MCI	SCI	NC	MCI	SCI	NC	MCI	SCI
N	8:1	10:6	10:6	9	16	16	9	16	16
M				65.0	69.6	75.9	28.2	23.3	10.8
Std. Deviation				11.5	7.3	10.5	1.1	1.7	5.0
Minimum				44.0	57.0	62.0	27.0	20.0	4.0
Maximum				77.0	80.0	96.0	30.0	25.0	18.0
X^2^	2.3			60.6			82.0 ***		

*** = *p* < 0.001.

**Table 2 diagnostics-13-03679-t002:** Dunn’s post hoc comparison of the MoCA score and WB FA between the SCI-MCI, SCI-NC, and MCI-NC groups.

Comparison	z	Wj	Wj′	*p*	pBonf	pHolm
SCI-MCI	−2.720	16.658	28.063	0.007 **	0.02 *	0.02 *
SCI-NC	0.099	16.656	16.167	0.921	1.000	0.921
MCI-NC	2.407	28.063	16.167	0.016 *	0.048 *	0.032 *

* *p* < 0.05, ** *p* < 0.01.

**Table 3 diagnostics-13-03679-t003:** Dunn’s post hoc comparison of the MoCA score and WB FA normative percentiles between the SCI-MCI, SCI-NC, and MCI-NC groups.

Comparison	z	Wj	Wj′	*p*	pBonf	pHolm
SCI-MCI	−2.361	18.438	28.438	0.018 *	0.055	0.036 *
SCI-NC	1.223	18.438	12.333	0.221	0.664	0.221
MCI-NC	3.227	28.438	12.333	0.001 **	0.004 **	0.004 **

* *p* < 0.05, ** *p* < 0.01.

## Data Availability

The datasets presented in this article are not readily available since they could have potentially identifiable data. Requests to access the datasets should be directed to nauris.zdanovskis@rsu.lv.

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
