# Peer review of "Whole Brain and Corpus Callosum Fractional Anisotropy Differences in Patients with Cognitive Impairment"

_diagnostics, 2023, doi:10.3390/diagnostics13243679_

Round 1

Reviewer 1 Report

Comments and Suggestions for Authors

The authors performed a DTI study of fractional anisotropy in MCI and demented subjects. Although the topic is interesting, there are many shortcomings in the article.  

1. They present DTI as a MRI technique able to study cognitive impairment (CI) but do not provide strong experimental evidence.

2. Inclusion and exclusion criteria need clarification (please describe clearly). Which diagnostic criteria were used? 

3. The authors used MoCA to diagnose MCI and dementia, but the such test is not able to distinguish between the two conditions.  MCI diagnosis is based on patient autonomy, was it evaluated?

4. MRI processing is not clear: preprocessing? Statistics analysis?

5. Paragraph 2.4 should provide a description of how each statistical test was utilized.

6. line 199:  "By performing Dunn post-hoc TEST statistically significant difference were found between SCI-MCI group...". Which difference? What do you mean by 'SCI-MCI group'? This is true for other groups as well.

The article's unclearness makes it difficult to evaluate the discussion 

Author Response

Dear reviewer,

Thank you very much for taking the time to review our manuscript. We appreciate Your insights and please find the detailed responses below

  1. “They present DTI as a MRI technique able to study cognitive impairment (CI) but do not provide strong experimental evidence.)”

The research using DTI to study cognitive impairment is ongoing and still maturing. There are new findings continuously emerging - nowadays with more complex multishell and multidirectional acquisition protocols. At the moment there are no clear statements on whether whole-brain FA is a viable early biomarker for cognitive impairment diagnostics. Our main goal was to assess whether whole-brain FA data from MR sequences that could be used on a daily basis (32 directions and single-shell) could be useful in early cognitive impairment detection.

  1. Inclusion and exclusion criteria need clarification (please describe clearly). Which diagnostic criteria were used?

We added more information In section “2.1. Selection of participants” where we describe exclusion criteria as follows:

  • clinical - participants were excluded from the study if they had clinically significant neurological disorders (such as history of tumours, severe strokes, vascular malformations, major depression, Parkinsons disease etc.), history of drug or alcohol abuse, diabetes mellitus, uncontrolled hypertension.
  • radiological - None of the participants had signs of cerebral amyloid angiopathy, more than 4 microbleeds, inta-/extra-axial tumors, vascular malformations or signs of other neurodegenerative disease,
  1. The authors used MoCA to diagnose MCI and dementia, but the such test is not able to distinguish between the two conditions. MCI diagnosis is based on patient autonomy, was it evaluated?

We did not use MoCA to diagnose MCI and dementia as MoCA is unable to assess patients based on dementia criteria. We used MoCA scale to assess severity of cognitive impairment and subdivided severity of cognitive impairment in 3 groups:

  • Normal cognition (NC) group (participants with MoCA scores ≥ 26);
  • Mild cognitive impairment (MCI) group (participants with MoCA ≥ 20 and ≤ 25);
  • Severe cognitive impairment (SCI) group (participants with MoCA≤ 19).

We changed the style of sentences in “Materials and Methods” section so that it would be clearer.

  1. MRI processing is not clear: preprocessing? Statistics analysis?

MR DTI processing was done by using Icometrix Software package and detailed processing steps and how estimation with the control group is done is described in prior publications by Timmermans et al (https://pubmed.ncbi.nlm.nih.gov/30605253/). These steps include eddy current correction with affine registration, diffusion tensor estimation using weighted linear least squares, FA map and distribution calculations and  region of intereset value extraction (whole brain and corpus callosum). We added information in the section “2.3 MRI acquisition and fractional anisotropy calculation.”

  1. Paragraph 2.4 should provide a description of how each statistical test was utilized.

Thank You for the suggestion! More detailed description was added to the 2.4. section.

  1. line 199: "By performing Dunn post-hoc TEST statistically significant difference were found between SCI-MCI group...". Which difference? What do you mean by 'SCI-MCI group'? This is true for other groups as well.

First we did Kruskall-Wallis test to identify whether there are statistically significant differences between the groups and there were statistically significant differences. Further, we did Dunn post-hoc test to identify between which groups these statistically significant differences were found, and we found that these differences were between severe cognitive impairment (SCI) group and mild cognitive impairment group (MCI).

Thank you for your time and suggestions!

Have a nice day!

Best regards,

Kalvis Kaļva

Reviewer 2 Report

Comments and Suggestions for Authors

The study focuses on DTI analysis of whole brain and corpus callosum as markers of cognitive impairment. The weakest point of the study is the poor description of the participants. The authors should add more information about the patients. Athors should to add information if the exclusion criteria were psychiatric diagnoses such as schizophrenic spectrum disorders, bipolar disorder. Where were the patients recruited from? Were they volunteers who were examined by a neurologist or patients from the hospital? Were patients with cognitive impairment treated with any medication or were they passed cognitive training?

Why 3 different images of the data in Figure 2-5? The authors can combine box plot and  jitter plot in the one figure.

If statistical differences in the corpus collosum were not found, then it makes no sense to indicate differences in mean values and pointed out that some group had higher value than the other. The authors showed the wrong conclusion. Maybe other regions of the brain should be studied, not only that whole brain?

Author Response

Dear reviewer,

Thank you very much for taking the time to review our manuscript. We appreciate Your insights and please find the detailed responses below

  1. The authors should add more information about the patients. Athors should add information if the exclusion criteria were psychiatric diagnoses such as schizophrenic spectrum disorders, bipolar disorder.

Yes, such psychiatric diagnoses were exclusion criteria. We added additional exclusion criteria that were used in section “2.1. Selection of participants”. In general, any clinically significant diseases that could affect cognitive scores were excluded. If in doubt, neurologist didn’t recruit the patient in study.

  1. Where were the patients recruited from? Were they volunteers who were examined by a neurologist or patients from the hospital?

The participants included in our study were referred to a neurologist based on their subjective complaints of cognitive impairment or suspected cognitive impairment based on primary physician assessment or family members.

  1. Were patients with cognitive impairment treated with any medication or were they passed cognitive training?

This is a cross-sectional study and patients presented in neurologists office previously were not diagnosed or treated with any medication related to cognitive impairment. Also, at the time of MRI they haven’t had any cognitive training or other interventions that could directly affect cognitive performance.

  1. Why 3 different images of the data in Figure 2-5? The authors can combine box plot and jitter plot in the one figure.

We have considered the suggestion to combine the scatter plot and the box plot into one figure. However, we have opted to keep them separate for the following reasons:

  1. Clarity of Individual Data Points: The jitter plot is designed to show every individual data point, which we feel is necessary for the transparency of data distribution and to highlight the variance within each group.
  2. Statistical Summary Visibility: The box plot provides a clear summary of the central tendency and dispersion, which may be obscured if overlaid directly with the jitter plot due to the density of data points.
  3. Focus on Outliers: Separating the plots allows us to emphasize outliers, which taking in account small sample size in our study could be of particular interest. These are more readily identifiable in a standalone scatter plot.

Separate jitter and box plots are not uncommon in the literature. We believe that maintaining separate plots for this data offers the most effective way to communicate our findings to the reader without compromising the detail or the interpretability of the results.

  1. If statistical differences in the corpus collosum were not found, then it makes no sense to indicate differences in mean values and pointed out that some group had higher value than the other. The authors showed the wrong conclusion.

We acknowledge the reviewer's point regarding the interpretation of the mean differences in the corpus callosum. The discussion was intended to describe observed patterns in the data, not to imply statistical significance where there is none. These should be considered as preliminary observations that may inform future hypothesis-driven research.

              In conclusions we clearly state that there were no statistically significant differences - We did not observe statistically significant differences between groups analysing CC FA and CC FA normative percentiles.”

  1. Maybe other regions of the brain should be studied, not only that whole brain?

Indeed! We completely agree that a focused analysis on specific brain tracts will yield additional insights. Our current study's scope was to assess general FA measures - whole-brain and corpus callosum. However, region specific analyses come with their own set of challenges, including the complexities of precise anatomical delineation, the need for additional processing techniques to accurately extract and analyze individual tracts, and the increased computational resources required for such calculations. These considerations, while beyond the scope of the present study, are indeed areas of interest for our ongoing research efforts.

Thank you for your time and suggestions!

Have a nice day!

Best regards,

Kalvis Kaļva

Round 2

Reviewer 1 Report

Comments and Suggestions for Authors

1. if DTI research in cognitive impairment is  ongoing, please modify your statement and provide references

2. it is still unclear how patients were diagnosed (which criteria? what diagnose? Alzheimer's Disease dementia? MCI due to AD?)

3. Moca is unable to distinguish MCI from dementia or NC (amnestic MCI patients can score above 26). Once again, how were diagnoses made? such point is crucial.

6. It is essential to specify which is higher or lower.

I'm OK with the other points

Author Response

Dear reviewer,

Thank you again for taking the time to review our manuscript. We appreciate Your insights and please find the detailed responses below:

  1. “If DTI research in cognitive impairment is ongoing, please modify your statement and provide references”

We added paragraph in introduction to emphasize this even more.

  1. “It is still unclear how patients were diagnosed (which criteria? what diagnose? Alzheimer's Disease dementia? MCI due to AD?)”

The participants included in our study were referred to a neurologist based on their subjective complaints of cognitive impairment or suspected cognitive impairment based on primary physician assessment.

Next, neurologist performed MoCA test and patient was assigned to group based on MoCA results. It is important to clarify that initial patient evaluation did not always result in definitive diagnoses of specific conditions like Mild Cognitive Impairment (MCI) or Alzheimer's Disease (AD) dementia. Diagnosis is a dynamic process, involving multiple assessments over time.

3.”Moca is unable to distinguish MCI from dementia or NC (amnestic MCI patients can score above 26). Once again, how were diagnoses made? such point is crucial.”

We completely agree and acknowledge the limitation of the MoCA in distinctly categorizing Mild Cognitive Impairment (MCI), dementia, and normal cognition (NC). Our study utilized MoCA scores primarily as a tool for stratifying cognitive performance, rather than as a definitive diagnostic tool for clinical syndromes or conditions classified under ICD.

To better reflect this methodology and avoid confusion, we could rename the groups as:

  • Optimal Cognitive Performance Group: Participants with MoCA scores of 26 or higher.
  • Moderate Performance Group: Participants with MoCA scores between 20 and 25, inclusive.
  • Low Performance Group: Participants with MoCA scores of 19 or lower.

We believe that retaining the original group names based on MoCA scores is appropriate for our study, as it directly reflects the methodology we employed and is described in “Materials and Methos”.

However, we are open to Your perspective on this issue. If You advise that renaming the groups for clarity would be beneficial, we are willing to implement these changes. We seek to balance accuracy in reflecting our methodology with clear and effective communication of our findings and would value Your input on this matter.

  1. It is essential to specify which is higher or lower.

We specified that the highest mean corpus callosum FA was found in the MCI patient group (mean value in group 0.628), but the lowest in SCI group (0.617); and highest mean CC FA normative percentiles were for MCI group (61.962), but the lowest mean percentiles for SCI group (55.262). Also these findings are reflected in figures.

Thank you for your time and suggestions!

Have a nice day!

Best regards,

Kalvis Kaļva

Reviewer 2 Report

Comments and Suggestions for Authors

It is still unclear why the study is retrospective. The authors write about it in the materials and methods section. Did the authors not recruit patients specifically for this study? If so, then they should specify the time period in which they recruited subjects.

Author Response

Dear reviewer,

Thank you again for taking the time to review our manuscript. We appreciate the insights and please find the detailed responses below:

  1. It is still unclear why the study is retrospective. The authors write about it in the materials and methods section. Did the authors not recruit patients specifically for this study? If so, then they should specify the time period in which they recruited subjects”

The study design is a retrospective analysis due to the specific research objectives (fractional anisotropy measures) and the availability of data. We did not actively recruit patients specifically for the fractional anisotropy measurements; instead, we utilized existing MRI data from patients who had previously undergone MR including DTI sequences as part of their diagnostic evaluation for suspected cognitive impairment.

The cognitive testing and MRI data was obtained in time period from January 2020 to December 2022 ensuring consistent diagnostic approach and same imaging protocol.

Nevertheless, we acknowledge the importance of clearly specifying this in our manuscript and updated the “Materials and Methods” section to include the time period of data collection, thereby providing a clearer context for our study design.

Thank you for your time and suggestions!

Have a nice day!

Best regards,

Kalvis Kaļva
